# Forest Resource Management and Its Climate-Change Mitigation Policies in Taiwan

**Wen-Tien Tsai**

Graduate Institute of Bioresources, National Pingtung University of Science and Technology, Pingtung 912, Taiwan; wttsai@mail.npust.edu.tw; Tel.: +886-8-7703202

**Abstract:** Based on high carbon emissions in recent years (i.e., about 11 metric tons in 2018) per capita in terms of carbon dioxide equivalents, Taiwan has actively development greenhouse gas (GHG) reduction action plans. One of the action plans has been to promote afforestation and reforestation in non-forested lands for carbon sequestration. Thus, this paper aims to address the forest resources in Taiwan by using the latest national survey, reporting on an interactive analysis of forest carbon sequestration, GHG emissions, and climate-change mitigation policies. In this regard, the methodology is based on the official websites of forest resources, GHG emissions, and carbon sequestration from the yearbooks, national statistics, and regulations relevant to the mitigation policies in the forestry sector. It is found that Taiwan's forest area is estimated to be 2.197 million hectares, which corresponds to a total forest stock volume of about 502.0 million cubic meters. During the period of 1990–2018, the change in total carbon sequestration did not vary much (with the exception of 2009), decreasing from 23.4 million metric tons in 1990 to 21.4 million metric tons in 2018. Compared to the total carbon dioxide emissions (i.e., 102.4 million metric tons in 1990 and 282.8 million metric tons in 2018), the contribution to GHG mitigation in the forestry sector shows a declining trend. However, biomass (i.e., wood) carbon sequestration indicates a slight increase from 20.4 million metric tons in 2010 to 20.7 million metric tons in 2018 due to the afforestation policy. Obviously, regulatory policies, based on the Forestry Act and the Greenhouse Gas Reduction & Management Act in 2015, play a vital role in mitigating GHG emissions in Taiwan. The discussion on the regulations is further addressed to highlight climate-change mitigation policies in Taiwan's forestry sector.

**Keywords:** greenhouse gas emission; forestry sector; carbon sequestration; climate-change mitigation; Taiwan

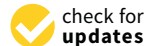

## 1. Introduction

Over the past three decades, there has been continuing concern about global warming, which is triggered by the emission of greenhouse gases (GHG) from anthropogenic activities, including the energy, industrial processes, agriculture, and waste management sectors. Consequently, the occurrence of extreme climate events has increased in frequency, intensity, spatial extent, duration, and timing, making weather patterns unpredictable. According to the Kyoto Protocol in 1997, the main GHG include carbon dioxide ($CO_2$), methane ($CH_4$), nitrous oxide ($N_2O$), and fluorinated gases (F-gases). At a global scale, these anthropogenic activities emit about 50 billion metric tons in terms of carbon dioxide equivalent ($CO_{2eq}$) each year [1], which includes the energy (electricity, heat, and transport) sector (73.2%), direct industrial processes (e.g., cement and chemicals and petrochemicals) sector (5.2%), waste management (i.e., wastewater treatment and sanitary landfill) sector (3.2%), and agriculture, forestry, and other land use (AFOLU) sector (18.4%).

To mitigate atmospheric concentrations of GHG that result in global warming and climate change, the pursuit of natural climate solutions has become a major focus of climate policy [2]. In this regard, land (or forests) and ocean ecosystems play a major role

in sequestrating atmospheric carbon. Regarding the removal of GHG from the forestry sector, this means carbon sequestration by forest resources, including trees and soils [3]. Although $CH_4$ and $N_2O$ can be emitted from ecosystems as a by-product of nitrification/denitrification and anaerobic decomposition, respectively [3], the key GHG of concern herein is $CO_2$. Its flux between the atmosphere and ecosystems is primarily controlled by uptake (sink) through plant photosynthesis and release via plant respiration, decomposition (of dead wood) and combustion of organic matter (e.g., forest fires). In general, there are three classes of forest-related activities affecting GHG concentration in the atmosphere [4,5], contributing to climate-change mitigation by carbon sink (or sequestration). Thus, the authors analyzed forest management models based on carbon conservation, carbon storage, and carbon substitution. The first is the establishment, enhancement, or protection of forest resources (or ecosystems) via afforestation and reforestation of non-forested lands, thus increasing and keeping carbon sequestered. The second is enhanced use of forest products by using wood in buildings (those of green building materials) and other long-lived objects. In other words, substituting wood-based products for energy-intensive materials such as brick, aluminum, or steel may reduce significant emissions of GHG from the use of fossil fuels. The third is the sustainable production of wood fuels (those from renewable energy sources), thinned wood [6], and wood-derived products from forests [7]. Although the burning of wood fuels and wood-derived waste releases $CO_2$ into the atmosphere, the regrowth of a sustainably managed forest will offset that release without a net contribution to GHG levels [8]. However, there is a continuing debate concerning woody bioenergy and the benefits of climate mitigation [9].

According to the guidelines developed by the International Panel on Climate Change (IPCC) [3], each country (or party) should submit a National Inventory Report (NIR) in response to climate-change mitigation and adaptation. Although Taiwan is not part of United Nations Framework Convention on Climate Change (UNFCCC), it has long been committed to fulfilling its responsibility as a member. Since 1998, Taiwan has taken initiatives to prepare the NIR. Starting from 2015, the central competent authority (i.e., Environmental Protection Administration (EPA)) submitted the NIR each year under the Greenhouse Gas Reduction and Management Act of 2015 [10]. The database not only provides an updated overview of the statistics of GHG emissions and sinks, but also reflects trends since 1990. Based on the updated NIR [11], the net GHG emissions (excluding the emissions from land use, land use change, and forestry) in Taiwan, contributing about 0.53% of the global GHG emissions, has grown greatly from $113{,}373 \times 10^3$ metric tons of $CO_{2eq}$ in 1990 to $275{,}039 \times 10^3$ metric tons of $CO_{2eq}$ in 2018. On the other hand, this increasing trend can be divided into two distinct periods. During the period of 1990–2005, it showed an annual growth as high as 5.90%. By contrast, an average annual growth of 0.21% appeared during the period of 2005–2018, reflecting the progressive development of renewable energy, energy saving, afforestation, and GHG mitigation by Taiwan government [12–18].

Regarding the regulatory and promotional measures for the analysis of carbon sequestration and its climate-change mitigation policies in Taiwan's forestry sector, very little discussion on the issue has been addressed in the literature. In a previous study [19], $CO_2$ sequestration by the forestry sector was addressed according to the "2015 Taiwan Greenhouse Gas Inventory" but did not discuss the correlations between forest resources and carbon sinks, and the regulatory policies for mitigating GHG emissions based on relevant laws. Therefore, the present paper consists of three key issues. First, information about forest resources in Taiwan is addressed by using the updated national survey [20]. Second, trend analysis of carbon sequestration in Taiwan's forestry sector is further performed. Finally, regulatory policies for enhancing carbon sequestration in the forestry sector are compiled to be in accordance with the Forestry Act and the Greenhouse Gas Reduction and Management Act.

## 2. Data Mining and Methodology

As described above, the main purposes of this study were to analyze forest resources and relevant carbon sequestration from the forestry sector since 1990. Using these trend

variations, the Sustainable Development Goals in Taiwan, and the National Climate Change Action Guidelines, regulatory policies were further connected with the Taiwan government efforts to mitigate GHG emissions from the forestry sector. Statistical data and regulatory policies are briefly summarized below.

- Forest resources in Taiwan

    The updated data on the statistics and status of forest resources in Taiwan were obtained from the relevant literature [21], official yearbook [22], and the website of the central competent agency (i.e., Forestry Bureau) [23].

- Inventory of GHG emissions from all sectors and carbon sequestration from the forestry sector

    Taiwan's statistics on GHG emissions and carbon sequestration from all sectors were compiled from the central competent authority (i.e., EPA) based on the methodology adopted by the IPCC [3]. Table 1 lists the relevant coefficients for estimating change in forest carbon stock in Taiwan [11]. In this work, the updated National Inventory Report was used to analyze the trends of total GHG emissions and carbon sequestration since 1990 [11].

**Table 1.** Local coefficients for estimating change in forest carbon stock in Taiwan [1].

| Forest Type | Wood Density | Biomass Expansion Factor | Biomass Conversion and Expansion Factor | Ratio [2] | Carbon Fraction | Annual Growth (m³/Hectare) |
|---|---|---|---|---|---|---|
| Natural coniferous | 0.41 | 1.27 | 0.51 | 0.22 | 0.4821 | 4.14 |
| Natural coniferous–broadleaf | 0.49 | 1.34 | 0.72 | 0.23 | 0.4756 | 10.05 |
| Natural broadleaf | 0.56 | 1.40 | 0.92 | 0.24 | 0.4691 | 3.58 |
| Artificial coniferous | 0.41 | 1.27 | 0.51 | 0.22 | 0.4821 | 8.11 |
| Artificial coniferous–broadleaf | 0.49 | 1.34 | 0.72 | 0.23 | 0.4756 | 10.37 |
| Artificial broadleaf | 0.56 | 1.40 | 0.92 | 0.24 | 0.4691 | 4.34 |
| Mixed bamboo | 0.49 | 1.34 | 0.72 | 0.23 | 0.4756 | 3.31 |
| Bamboo | 0.62 | 1.40 | - | 0.46 | 0.4732 | 13.84 [3] |

[1] Source [11]. [2] Ratio of below-ground biomass to above-ground biomass for a specific forest type. [3] Unit: metric ton/hectare.

- Regulatory policies for mitigating GHG emissions from the forestry sector

    The relevant information about the regulatory policies for mitigating emissions of GHG from the forestry sector was based on the laws and national guidelines by accessing the official website [10]. The most important laws include the Forestry Act and the Greenhouse Gas Reduction and Management Act. The former refers to preservation of forest resources, natural functions of forests, and their economic viability executed by the Council of Agriculture. The latter is responsible for the establishment of a green low-carbon society, including renewable energy (e.g., wood fuels as biomass energy), low-carbon products, green building materials (e.g., wood–based materials), afforestation (and reforestation), and so on.

## 3. Results

### 3.1. Trend Analysis of Forest Rresources in Taiwan

As Taiwan lies on the border between a tropical and subtropical climate (i.e., the Tropic of Cancer), its forests can be classified as tropical forests, subtropical forests, or temperate forests based on their altitude, climate, and humidity. According to the 4th National Forest Resource Investigation [20], performed by the Forestry Bureau and completed in 2014, Taiwan's forest area is estimated to be 2.197 million hectares, which corresponds to forest coverage of 60.71% and 0.093 hectare per capita. In terms of forest type, Figure 1 shows the dominant types and their percentages, which are summarized below:

- Broadleaf forests: 1470 thousand hectares (estimated) and 67% of forest area.

- Coniferous forests: 299 thousand hectares (estimated) and 14% of forest area.
- Mixed coniferous and broadleaf forests: 171 thousand hectares (estimated) and 8% of forest area.
- Bamboo forests: 113 thousand hectares (estimated) and 5% of forest area.
- Mixed bamboo forests: 115 thousand hectares (estimated) and 5% of forest area.
- Pending forests: 29 thousand hectares (estimated) and 1% of forest area.

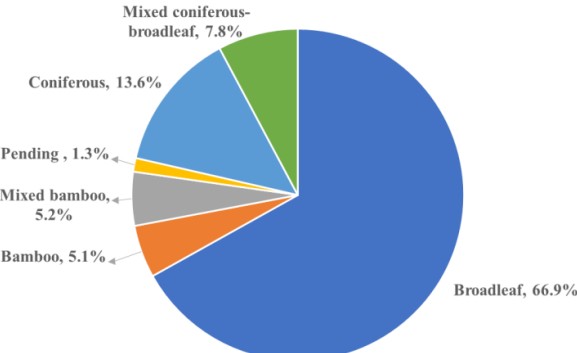

**Figure 1.** Forest types and their percentages based on the total forest area in Taiwan [11].

The total forest stock volume is estimated at 502.0 million cubic meters, calculated by area of forest type and its stock volume per hectare on average. Table 2 lists the forest types and their stock volumes [11], showing that stock volume and percentage in natural forests accounts for 429.5 million cubic meters (including 103.1 million cubic meters in coniferous forests, 268.9 million cubic meters in broadleaf forests and 57.6 million cubic meters in mixed coniferous–broadleaf forests) and 85.6%, respectively. By contrast, the stock volume and percentage in artificial forests (forest plantation) occupied 64.5 million cubic meters (including 29.5 million cubic meters in coniferous forests, 19.8 million cubic meters in broadleaf forests and 15.2 million cubic meters in the mixed coniferous–broadleaf forests) and 12.8%, respectively. The remaining belongs to mixed bamboo forests, which only share 8.0 million cubic meters and 1.6%, respectively. Furthermore, the total carbon storage (as carbon dioxide) can be obtained by using the IPCC method [3], showing an estimated value of about 754.2 million metric tons. In terms of forest type, the carbon storages (percentages) are summarized below:

- Broadleaf forests: 469.0 million metric tons (62.2%).
- Coniferous forests: 156.3 million metric tons (20.7%).
- Mixed coniferous and broadleaf forests: 103.5 million metric tons (13.7%).
- Bamboo forests: 14.6 million metric tons (1.9%).
- Mixed bamboo forests: 10.8 million metric tons (1.4%).

**Table 2.** The forest types and their stock volumes in Taiwan [1].

| Forest Type | Stock Volume ($10^3$ m$^3$) |
| --- | --- |
| Natural forest | 429,525 |
| Coniferous forest | 103,070 |
| White fir | 11,562 |
| Hemlock | 44,834 |
| Spruce | 6191 |
| Cypress | 22,189 |
| Pine | 17,176 |
| Others | 1118 |
| Broadleaf forest | 268,874 |
| Mixed coniferous–broadleaf forest | 57,581 |

**Table 2.** *Cont.*

| Forest Type | Stock Volume ($10^3$ $m^3$) |
|:---:|:---:|
| Artificial forest | 64,491 |
| Coniferous forest | 29,474 |
| Cypress | 5740 |
| Pine | 7940 |
| Chinese fir | 1658 |
| Willow | 11,882 |
| Others | 2254 |
| Broadleaf forest | 19,790 |
| Mixed coniferous–broadleaf forest | 15,226 |
| Mixed bamboo forest | 8018 |
| Total | 502,034 |

[1] Source [11].

*3.2. Trend Analysis of Carbon Sequestration in Taiwan's Forestry Sector*

In the forestry sector, the most important GHG is $CO_2$. Its fluxes between the atmosphere and ecosystems are primarily controlled by uptake through plant photosynthesis and releases via respiration, decomposition, and combustion of organic matter. Regarding the other GHG, such as nitrous oxide ($N_2O$) and methane ($CH_4$), they could be emitted from ecosystems via forest fire, and nitrification, denitrification, or anaerobic decomposition of organic matters (e.g., dead wood and soils), but these sources are relatively small and are not accounted for in the work. Table 3 lists the variations on the carbon sequestration in Taiwan's forestry sector since 1990 [11], including the sources of biomass carbon sequestration and biomass carbon emission. Regarding the biomass carbon emission sources, they may include wood removal (harvest or thinning), fuel wood, and disturbance (e.g., typhoon, forest fire, illegal logging, and reclamation). For example, the typhoon Morakot in August 2009 triggered severe disasters (e.g., landslides) in Central and Southern Taiwan due to amazing rainfall (i.e., 2965 mm in 4 days) [24], producing about 1.25 million metric tons of driftwood and thus causing large losses in volume of wood and forest land. Consequently, the total carbon sequestration in 2009 was below 21 million metric tons (i.e., 18.9 million metric tons), which was attributed to the reduction in forest land area and increase in the burning/combustion of driftwood and other biomass residues. Based on the data in Table 3, the important features are briefly addressed as follows:

1. During the period 1990–2018, the change in the total carbon sequestration did not vary much (with the exception in 2009) ranging from 21.4 to 23.4 million metric tons. This stable situation was mainly due to the dynamic balance between increased sequestration from the annual growth of forest resources and decreased sequestration from falling and felling trees.
2. Total carbon sequestration in 1990 was 23.4 million metric tons. By contrast, the value in 2018 dropped to 21.5 million metric tons, thus indicating an annually decline rate of 0.30%. Compared to the value (i.e., 21.9 million metric tons) in 2005 (base year), sequestration in 2018 was down by 1.88%.
3. In recent years (2010–2018), biomass (i.e., wood) carbon sequestration indicated a slight increase from 20.4 million metric tons in 2010 to 20.7 million metric tons in 2018. This increase should be attributed to afforestation and forest management policies during this period. According to official statistics [22], forest land area increased from 2,101,719 hectares in 2010 to 2,197,090 hectares in 2018.

Apart from the forestry (and land use change) sector in carbon sequestration, the GHG emission sources come from four sectors—energy, industrial processes and product uses, agriculture, and waste. Table 4 summarizes the data of $CO_2$ emissions and sinks since 1990 [11]. It shows that total $CO_2$ emissions in Taiwan increased from 124.1 million metric tons in 1990 to 282.8 million metric tons in 2018, which is equivalent to an average increase rate of 3.0% annually. Among these sources, emission from the energy sector were more

than 90%. By subtracting carbon sequestration (sink) in the forestry sector as discussed in the previous paragraph, the net $CO_2$ emissions in Taiwan indicate a consistent trend, increasing from 100.7 million metric tons in 1990 to 261.3 million metric tons in 2013, with $CO_2$ emissions increasing by about 160%. Due to the slight incline in carbon sequestration by the forestry sector (as seen in Table 3), an average annual growth rate of 3.5% was obtained from the data on the net $CO_2$ emissions during the period of 1990–2018.

**Table 3.** Taiwan's carbon sequestration by forestry sector [1].

| Sequestration/Emission | | Year | | | | | | | | |
|---|---|---|---|---|---|---|---|---|---|---|
| | | **1990** | **1995** | **2000** | **2005** | **2010** | **2015** | **2016** | **2017** | **2018** |
| Forests maintaining forests | Biomass carbon sequestration | 23,902 | 23,146 | 22,201 | 21,255 | 20,392 | 20,546 | 20,575 | 20,612 | 20,656 |
| | Biomass carbon emission | 607 | 202 | 389 | 369 | 218 | 189 | 153 | 111 | 83 |
| Other lands turned to forests | Biomass carbon sequestration | 91 | 288 | 665 | 1032 | 1240 | 1068 | 1029 | 981 | 934 |
| Total carbon sequestration [2] | | 23,386 | 23,233 | 22,476 | 22,918 | 22,413 | 21,425 | 21,451 | 21,482 | 21,507 |

[1] Source [11]; unit: $10^3$ metric tons. [2] The values were obtained by accumulating the biomass carbon sequestration and then subtracting the biomass carbon emission.

**Table 4.** Taiwan's carbon dioxide emissions and sinks [1].

| Emission/Sink | Year | | | | | | | | |
|---|---|---|---|---|---|---|---|---|---|
| | **1990** | **1995** | **2000** | **2005** | **2010** | **2015** | **2016** | **2017** | **2018** |
| Total emission | 124,066 | 168,873 | 226,978 | 266,460 | 270,148 | 275,835 | 279,705 | 284,812 | 282,842 |
| Sink by forestry sector | 23,386 | 23,233 | 22,476 | 22,918 | 22,413 | 21,425 | 21,451 | 21,482 | 21,507 |
| Net emission | 100,680 | 145,640 | 204,502 | 244,542 | 248,735 | 254,410 | 258,254 | 263,330 | 261,335 |

[1] Source [11]; unit: $10^3$ metric tons.

## 4. Discussion

As described above, the forestry sector plays a critical role in climate-change mitigating by sequestering GHG (i.e., carbon dioxide) in the atmospheric environment. Using IPCC methods and the central government agency (i.e., Forestry Bureau) in Taiwan, the contribution to GHG absorption by the forestry sector in Taiwan is about 7% based on total GHG emissions. However, this work focuses on the aspects of the forestry sector from the point of view of regulatory measures or policies. It does not study other sectors (i.e., energy sector and waste management sector) leading to the contribution of GHG emission mitigation, because these sectors can reuse lignocellulosic residues (e.g., wood sawdust) as resources for the production of biomass energy and building materials. The obtained results indicate a need for further research to examine the net contribution of the forestry sector to the observed GHG emission trends. With respect to the policies and regulations for mitigating GHG emissions and sinks in Taiwan's forestry sector, the most relevant laws are based on the Forestry Act and the Greenhouse Gas Reduction and Management Act [10]. In this regard, it can be seen that forest land area shows a slightly increasing trend under the afforestation policy, as described in the Section 3.2. Furthermore, in line with the sustainable development goals (SDGs) launched by the United Nations in 2015 [25], the central competent agency (i.e., Forestry Bureau, Council of Agriculture) set the forest land area via afforestation and reforestation as one of the sustainable development targets (i.e., 3636 hectares by 2020, and 7080 hectares by 2030) in the 13th SDGs (Climate action: Taking urgent action to combat climate change and its impacts) [26]. In addition, many paper- and wood-derived products in Taiwan have been commended by the Forest Management (FM)

division of the Forest Stewardship Council (FSC) [27], showing that these products come from responsible forest management.

Based on the two acts, their mitigation policies and promotional measures can be briefly summarized and addressed as follows.

### 4.1. Forestry Act

The Act, which was revised in 2016, aimed to preserve forest resources, natural functions of forests and their economic viability, and also protect conservation-valuable trees and their growth habitats. Under the Act, carbon management with relevance to the forest resource can be further summarized as follows:

(1)　Carbon sequestration

- To increase carbon reserves in forest and its soils, the central government agency requests that the central land administrative authority classifies the undeveloped mountains and lands as forestland (Article 6).
- To maintain forest ecology and preserve biodiversity, a so-called "nature reserve" may be designated within a forest zone (Article 17-1).
- To encourage reforestation and award forest business by private individuals and/or organizations, the central government agency may provide free seedings, rewards, long-term low-interest loans, or other assistances and promotions. In addition, the competent agency will establish a reforestation fund for the above-mentioned issue (Article 47, Article 48, and Article 48-1).

(2)　Carbon conservation

- To conserve the carbon pool, a forest protection area may be designated in the forest periphery. Furthermore, prescribed burns and ignitions will not be permitted in the forest and forest protection zones (Article 33 and Article 34).
- To prevent forest fire, the government agency will establish a forest fire squad and organize a volunteer forest fire squad as needed (Article 35).
- To eliminate or prevent biological hazards or disturbances in the forest, the forest owner shall be responsible for them (Article 38).

(3)　Carbon substitution

- To increase the transfer of carbon from forest biomass to other wood products such as building materials and/or biomass fuels, the harvesting of national forest yields shall be carried out according to the annual logging plan (Article 15).
- To strengthen forest business, forestry practitioners may organize a forestry cooperative association based on the Cooperation Association Act (Article 19).

### 4.2. Greenhouse Gas Reduction and Management Act

In response to the Post-Kyoto Protocol on climate change, this Act, passed in July 2015, aims to reduce GHG emissions at all levels of central and local governments by establishing adaptation strategies. Under Article 4 in the Act, the core target is to reduce GHG emissions to less than 50% of 2005 levels by 2050. In the definition of carbon sinks by the Act, this means any process or mechanism that removes a GHG from the atmosphere, such as trees and forests. In accordance with the Act, the relevant policies for mitigating net GHG emissions from the forestry sector by carbon sinks or carbon management include the following items.

(1)　The relevant central government agencies will promote GHG reduction and climate-change adaptation through their actions, including

- Development of renewable energy and energy technology; i.e., the use of wood-based biomass energy.
- Reduction and management of GHG emissions from buildings; i.e., the use of woods.
- Waste recycling and use; i.e., the reuse of waste wood as fuels or materials.

- Forest resource management in the forest's carbon sequestration.
- Educational propaganda in climate-change adaptation and GHG reduction.

(2) The central competent authority (i.e., EPA) will develop a "National Climate Change Action Guideline" and "Greenhouse Gas Reduction Action Plan" based on the nation's economy, energy supplies, environments, international situation, and the assignment responsibilities by the relevant central government agencies [28,29].

- The "National Climate Change Action Guideline" will be reviewed and examined once every five years.
- The "Greenhouse Gas Reduction Action Plan" will contain regulatory items, including staged control goals, implementation timetables, implementation strategies, expected benefits, and an evaluation mechanism.
- According to the "Greenhouse Gas Reduction Action Plan", the relevant central government agencies will include GHG emission targets, timetables, and economic incentive measures.

(3) The central competent authority (i.e., EPA) will implement the cap-and-trade scheme based on the UNFCCC and its agreements, or relevant international conventions in response to international GHG reduction requirements.

- After implementing GHG emission/sink accounting, verification, and registration as well as establishment of the regulations of allocation, offset, auction, sale, and allowance trading, the central competent authority in consultation with the central industrial competent authorities will implement the cap-and-trade scheme.
- Major GHG emission sources may procure emissions allowances from the GHG emission reduction projects (e.g., offset project), which have been validated and registered in the holding account.
- To reduce carbon footprint, the procurement of GHG reduction credits will give priority to domestic efforts.

## 5. Conclusions and Prospects

It was revealed that over 21 million metric tons of total carbon sequestration in Taiwan comes from the forestry sector based on a forest area of about 2.2 million hectares. In this paper, an interactive analysis of forest resources, carbon sequestration, greenhouse gas emissions, and climate-change mitigation policies in Taiwan's forestry sector was carried out. Although the total carbon sequestration during the period of 1990–2018 indicated a declining trend, the biomass (i.e., wood) carbon sequestration showed a slight increase from 20.4 million metric tons in 2010 to 20.7 million metric tons in 2018 due to the afforestation and forest protection policies. Total carbon dioxide emissions significantly increased from 102.4 million metric tons in 1990 to 282.8 million metric tons in 2018, meaning a declining contribution to GHG mitigation by the forestry sector. In line with SDGs launched by the Taiwan government in 2018, the central competent agency (i.e., Forestry Bureau, Council of Agriculture) set the forest land area via afforestation and reforestation as one of the sustainable development targets (i.e., 3636 hectares by 2020, and 7080 hectares by 2030) in the 13th SDGs (Climate action: Taking urgent action to combat climate change and its impacts). In the near future, Taiwan's forestry sector will play a more important role in climate-change mitigation under the authorization of the Greenhouse Gas Reduction and Management Act.

**Funding:** This research received no external funding.

**Institutional Review Board Statement:** Not applicable.

**Informed Consent Statement:** Not applicable.

**Conflicts of Interest:** The author declared that they have no known competing financial interests or personal relationships that could have appeared to influence the work reported in this paper.

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
