# Peer review of "Forest Resource Management and Its Climate-Change Mitigation Policies in Taiwan"

_climate, doi:10.3390/cli9010003_

Round 1
Reviewer 1 Report
well written and clear. no remarks to suggest
Author Response
Q1. Well written and clear. No remarks to suggest.
Reply: I thank the reviewer for his/her positive comment.
Reviewer 2 Report
Forest resource management and its climate change mitigation policies in Taiwan
The paper deals with an interesting issue that is contribution of forest resource management to climate change mitigation. Unfortunately, the work is not of a scientific nature.
The author did not formulate research questions for which he will give clear answers. In the paper, an expert opinion on the forest sector and its climate change mitigation policies in Taiwan is presented.
In part Introduction, generalized background of the topic is based mainly on political documents and strategies. To add some results of research papers would be very useful.
Poor selection of research methods means that the author could not conduct a good discussion about his results.
The part “Results and discussion” seems to be a description of the development of forestry in Taiwan and mitigation policies of Taiwan´s forestry sector than a discussion on research results.
Dividing this part into two separate parts (“Results” and “Discussion”) would be appropriate and would contribute to the higher quality of the article.
In the discussion part, it would be appropriate to describe the specific measures and facts that contributed to the carbon sequestration in Taiwan.
I suggest major adjustments of the article before its publishing.
Author Response
Q1. Introduction: Generalized background of the topic is based mainly on political documents and strategies. To add some results of research papers would be very useful.
Reply: The description about the results of research papers (two references added additionally) has been added to the Introduction (the second paragraph).
“To mitigate atmospheric concentrations of GHG that result in global warming and climate change, natural climate solutions have become a major focus of climate policy [2]. In this regard, land (or forests) and ocean ecosystems play a major role in sequestrating atmospheric carbon. Regarding the removal of GHG from the forestry sector, it means the carbon sequestration by forest resources, including trees and soils [3]. Although CH4 and N2O can be emitted from ecosystems as a by-product of nitrification/denitrification and anaerobic decomposition, respectively [3], the key GHG of concern herein is CO2. Its flux between the atmosphere and ecosystems is primarily controlled by uptake (sink) through plant photosynthesis and releases via plant respiration, decomposition (of dead wood) and combustion of organic matter (e.g., forest fire). In general, there are three classes of forest-related activities of affecting GHG concentration in the atmosphere [4, 5], contributing to the climate change mitigation by carbon sink (or sequestration). Thus, the authors analyzed the forest management models based on carbon conservation, carbon storage, and carbon substitution. The first is the establishment, enhancement or protection of forest resources (or ecosystems) via afforestation and reforestation of non-forested lands, thus increasing and keeping carbon sequestered. The second is the enhanced use of forest products by using wood in buildings (one of green building materials) and other long-lived objects. In other words, substituting wood-based products for energy-intensive materials such as brick, aluminum or steel may reduce the significant emissions of GHG from the use of fossil fuels. The third is sustainable production of wood fuels (one of renewable energy sources), thinned wood [6] and wood-derived products from forests [7]. Although burning of wood fuels and wood-derived waste releases CO2 into the atmosphere, the regrowth of a sustainably managed forest will offset that release without net contribution to GHG levels [8]. However, there is a continuing debate concerning woody bioenergy in the benefits of climate mitigation [9].”
Q2. Poor selection of research methods means that the author could not conduct a good discussion about his results.
Reply: I agree to the reviewer for this comment. In this work, the author focused on the data mining from the official database to analyze the carbon sequestration in the Taiwan’s forestry sector. In addition, these results further indicated the decoupling of greenhouse gas emissions from economic growth, which can attributed to the climate change mitigation policies adopted the Taiwan government in recent years.
Q3. Results and discussion: It seems to be a description of the development of forestry in Taiwan and mitigation policies of Taiwan´s forestry sector than a discussion on research results. Dividing this part into two separate parts (“Results” and “Discussion”) would be appropriate and would contribute to the higher quality of the article.
Reply: Dividing this part into two separate parts (“Results” and “Discussion”) as suggested by the reviewer.
Q4. In the discussion part, it would be appropriate to describe the specific measures and facts that contributed to the carbon sequestration in Taiwan.
Reply: As suggested by the reviewer, the description about the specific measures and facts that contributed to the carbon sequestration in Taiwan has been added to the Discussion. The limitation of the study was also addressed.
“As described above, the forestry sector plays a critical role in the climate change mitigating by sequestering GHG (i.e., carbon dioxide) in the atmospheric environment. Using the IPCC methods by the central government agency (i.e., Forestry Bureau) in Taiwan, the contribution to GHG absorption by forestry sector in Taiwan are about 7% based on total GHG emissions. However, this work focused on the aspects of the forestry sector from the regulatory measures or policies. It did not study other sectors (i.e., energy sector and waste management sector) leading to the contribution of GHG emission mitigation because these sectors could reuse lignocellulosic residues (e.g., wood sawdust) as resources for the productions of biomass energy and building materials. The obtained results indicated a necessary for further research in order to examine the net contribution of the forestry sector by the observed GHG emission trends. …….”

Round 2
Reviewer 2 Report
The changes made in the revised manuscript has improved the quality of the paper and my overall recommendation is therefore that the manuscript can be accepted for publication in its current form.